# Metformin Treatment Induces Different Response in Pheochromocytoma/Paraganglioma Tumour Cells and in Primary Fibroblasts

**DOI:** 10.3390/cancers14143471

**Published:** 2022-07-17

**Authors:** Serena Martinelli, Francesca Amore, Tommaso Mello, Massimo Mannelli, Mario Maggi, Elena Rapizzi

**Affiliations:** 1Department of Experimental and Clinical Biomedical Sciences “Mario Serio”, University of Florence, 50134 Florence, Italy; serena.martinelli@unifi.it (S.M.); francesca.amore@unifi.it (F.A.); tommaso.mello@unifi.it (T.M.); massimo.mannelli@unifi.it (M.M.); mario.maggi@unifi.it (M.M.); 2Centro di Ricerca e Innovazione Sulle Patologie Surrenaliche, AOU Careggi, 50134 Florence, Italy; 3ENS@T Center of Excellence, 50134 Florence, Italy; 4Department of Experimental and Clinical Medicine, University of Florence, 50134 Florence, Italy

**Keywords:** pheochromocytoma/paraganglioma, tumour microenvironment, metformin, tumour metabolism

## Abstract

**Simple Summary:**

Pheochromocytoma/paragangliomas (PPGLs) are neuroendocrine tumours and are often non-metastatic. However, no effective treatment is available for their metastatic form. Recent studies have shown that metformin exhibits antiproliferative activity in many human cancers, including PPGLs. Nevertheless, no data are available concerning whether metformin is also able to inhibit PPGL metastatic spread. A tumour is a very complex system, comprising not only cancer cells, but also other cells that all together form the so-called tumour microenvironment. Cancer-associated fibroblasts are residential or recruited fibroblasts, transformed by cancer cells, to promote tumour growth and spread. Therefore, the interplay between tumour cells and cancer-associated fibroblasts has become an interesting target for cancer therapy. Here, we demonstrate that metformin has different effects on cancer cells and fibroblasts, providing evidence that metformin may hold promise for altering tumour microenvironment homeostasis. Improving our knowledge on malignant tumour microenvironment properties could lead to develop complementary strategies to target tumour spread and progression.

**Abstract:**

Pheochromocytoma/paragangliomas (PPGLs) are neuroendocrine tumours, often non-metastatic, but without available effective treatment for their metastatic form. Recent studies have shown that metformin exhibits antiproliferative activity in many human cancers, including PPGLs. Nevertheless, no data are available on the role of metformin on PPGL cells (two-dimension, 2D) and spheroids (three-dimension, 3D) migration/invasion. In this study, we observed that metformin exerts an antiproliferative effect on 2D and 3D cultures of pheochromocytoma mouse tumour tissue (MTT), either silenced or not for the SDHB subunit. However, metformin did not affect MTT migration. On the other hand, metformin did not have a short-term effect on the proliferation of mouse primary fibroblasts, but significantly decreased their ability to migrate. Although the metabolic changes induced by metformin were similar between MTT and fibroblasts (i.e., an overall decrease of ATP production and an increase in intracellular lactate concentration) the activated signalling pathways were different. Indeed, after metformin administration, MTT showed a reduced phosphorylation of Akt and Erk1/2, while fibroblasts exhibited a downregulation of N-Cadherin and an upregulation of E-Cadherin. Herein, we demonstrated that metformin has different effects on cell growth and spread depending on the cell type nature, underlining the importance of the tumour microenvironment in dictating the drug response.

## 1. Introduction

Pheochromocytomas and paragangliomas (together PPGLs) arise from neural crest-derived chromaffin cells of the adrenal medulla or from the sympathetic or parasympathetic paraganglia, respectively. Over 40% of PPGLs are germline mutated in one of the 23 susceptibility genes identified so far, and the most frequently mutated are those encoding for the succinate dehydrogenase (SDH) subunits (SDHA, SDHB, SDHC, SDHD) [1,2,3]. Moreover, almost 20% of *SDHB* mutated patients develop metastatic lesions [4]. Indeed, along with extra-adrenal location, size of the tumour, and younger age, SDHB mutations are considered one of the risk factors leading to a malignant phenotype [5]. The first-choice treatment for localized disease is surgery. In contrast, for inoperable metastatic tumours, there is no definitive curative or standard medical therapy, and 40–80% of patients show an overall survival of five years from the diagnosis of the first metastasis [6,7].

Metformin (N, N-dimethylbiguanide) is the most prescribed antihyperglycemic oral drug worldwide [8]. Recently, metformin has been proposed as a potential antitumoral drug due to its inhibitory effect on growth and survival of different cell lines, including neuroblastoma [9], glioblastoma cells [10,11], and rat PPGL cells [12]. Its anticarcinogenic activity was also supported by evidence obtained from epidemiological studies and clinical trials on endometrial [13], colorectal [14], prostate [15], breast [16], adrenocortical [17], and head and neck cancers [18]. However, whether metformin plays a role in inhibiting the migration of PPGL cells has yet to be explored.

Tumour microenvironment (TME) is a very complex system, comprising not only cancer cells, but also macrophages [19,20], vascular endothelial cells [21,22], immune cells [23,24], and stromal cancer-associated fibroblasts (CAFs) [25,26]. CAFs are residential or recruited fibroblasts, often transformed by cancer cells, to promote tumour growth and spread. Indeed, the interplay between tumour cells and CAFs within TME has been recognized as a driving force in the progression of various tumours, including PPGLs [27,28]. Therefore, TME has become an interesting target for cancer therapy.

To date, it is well accepted that metformin mainly acts by inhibiting mitochondrial electron transport chain complex I, causing an imbalanced redox status, reducing energy production and biosynthesis, and disturbing cell metabolism [29,30]. This cellular metabolic stress culminates with AMP-activated protein kinase (AMPK) activation [31,32]. However, the relationship between functional and metabolic effects induced by metformin on different TME cell types remains unclear, and further studies to clarify this issue are needed.

In the present study, we investigated the effects of metformin on proliferation and migration of 2D and 3D cultures of mouse PPGL tumour cells and primary fibroblasts. Moreover, in the same cell populations we tried to identify the intracellular signalling pathways modulated by metformin.

## 2. Methods and Materials

### 2.1. Cell Cultures

Primary mouse fibroblasts, obtained from the whole peeled leg of new-born C57/BL by enzymatic digestion with collagenase I and trypsin, were cultured in complete Dulbecco’s Modified Eagle Medium (DMEM): DMEM with 2 mM L-Glutamine, 10% FCS, 100 U/mL penicillin, and 100 μg/mL streptomycin [33]. The pheochromocytoma mouse tumour tissue derived (MTT) cell lines, stably silenced or not for the SDHB subunit (MTT SDHB sil, and MTT wt, respectively), already available and used in our lab [33], were grown in complete DMEM supplemented with 5% horse serum (HS), and 1µg/mL puromycin. In both cases cells were grown at 37 °C in a 5% CO_2_ humidified atmosphere.

### 2.2. Spheroids Induction and Growth

To generate MTT spheroids, 5 × 10^3^ cells/well in cell-repellent surface, round bottomed 96-well plates (Greiner Bio-one, Kremsmünster, Austria), were centrifuged at 220× *g* at room temperature for 10 min. This procedure generated spheroids with homogeneous size and geometry. After 48 h spheroids showed a round morphology with clear margins and used for further experiments as T0 [27]. To assess spheroids growth, culture medium was replaced with fresh medium with or without 4 mM of metformin. Images of the spheroids were acquired at different time points (T0, 2, 3, and 6 days), and spheroid diameters were measured with ImageJ software [34].

### 2.3. Conditioned Medium

Mouse primary fibroblasts and MTT cells (wt or SDHB sil) were co-cultured in a 1:2 ratio. The cells were allowed to adhere overnight, then complete DMEM was replaced with a fresh one. After 72 h, the medium, now called “conditioned medium”, was collected, centrifuged for 5 min at 1200 rpm [27,33], and used for CAFs induction and confocal microscopy studies.

### 2.4. Proliferation Assay

Cells (MTT wt, MTT SDHB sil, and fibroblasts) were seeded at 2.5 × 10^4^/mL into 24 well plates, allowed to adhere overnight, and then treated with increasing metformin concentrations (0, 2, 4, 8 mM), in complete DMEM. Cells were counted by a haemocytometer after 1, 2, 3, and 6 days of treatment. To evaluate proliferation in acidic medium or in a non-acidic medium, cells were grown in complete DMEM at pH 7.4, or complete DMEM at pH 7.4 + 4 mM of metformin (Merck, Darmstadt, Germany), or complete DMEM acidified at pH 6.5 by adding HCl, or DMEM plus 20 mM of hepes, or DMEM plus 4 mM of metformin and 20 mM of hepes and then counted after 2, 3, and 6 days. 

To assess fibroblasts proliferation at longer times, cells were seeded at 5 × 10^3^/mL into 24 wells, treated with 2 mM or 4 mM of metformin, and then enumerated at day 1, 2, 3, 6 and 9.

### 2.5. Western Blot

Cells were lysed in buffer containing 50 mM Tris-HCl pH = 7.5, 120 mM NaCl, 1 mM EGTA, 6 mM EDTA, 15 mM Na_4_P_2_O_7_, 20 mM NaF, and 1% Triton X-100 protease inhibitor cocktail. Lysates were clarified by centrifugation at 10,000× *g* for 15 min at 4 °C, and supernatants were quantified for protein content (Coomassie Blue reagent, Bio-Rad, Hercules, CA, USA). All passages were carried out on ice as described by Rapizzi et al. (Rapizzi et al., 2014). Proteins were separated by 8% or 12% sodium dodecyl sulfate polyacrylamide (SDS/PAGE) gels and transferred to PVDF membranes (Fisher Scientific or Immobilon, Millipore, MA, USA). Primary antibodies against alpha-tubulin (#3873), cyclin D1 (#55506), phospho-AMPK (Thr172) (#2535), phospho-Erk1/2 (#9106s), Erk1/2 (#9102s), N-Cadherin (#14215s), and E-Cadherin (#14472s), were from Cell Signaling Technology. Phospho-Akt (Ser473) (sc7985), Akt (sc8312), and all the secondary HRP conjugated antibodies (anti-mouse, sc-2005, and anti-rabbit, sc-2004) were obtained from Santa Cruz Biotechnology (Dallas, Texas, USA).

Protein bands were detected with ECL reagents (Immobilon Crescendo, Millipore, Burlington, MA, USA). ImageJ software was used for the densitometric analysis of the bands.

The original whole blot (uncropped blots) showing all the bands are reported in the Appendix A.

### 2.6. pH Measurement

Cells were plated at 9 × 10^5^/p60 dishes in complete DMEM, allowed to adhere overnight, and then treated with increasing metformin concentrations (0, 2, 4, 8 mM). After 72 h, culture media were harvested, and their pH immediately measured by pHmeter (Basic 20 pH, Crison, Barcelona, Spain).

### 2.7. Intracellular Lactate and Adenosine Triphosphate (ATP) Concentration Measurement

Briefly, cells were seeded (2.5 × 10^4^/well) into 96 well plates, allow to adhere overnight, and then treated or not with 2 or 4 mM of metformin for 72 h. For lactate measurements, Lactate Colorimetric/Fluorometric Assay Kit (Biovision, Milpitas, CA, USA) was used according to the manufacturer’s protocol. Briefly, cells were washed twice in PBS, lysated, and 50 μL/each sample in duplicates were mixed with 50 μL of mix solution in a 96-well plate, and left at 37 °C for 30 min. The absorbance was measured at 570 nm wavelength using a microplate reader VICTOR^3^ 1420 Multilabel Counter (Packard Instruments, Perkin-Elmer, Waltham, MA, USA) and normalized on cell number.

For ATP levels, CellTiter-Glo luminescent cell viability assay (Promega) was used according to the manufacturer’s protocol. Cells were washed twice in PBS, and 125 μL of PBS plus 125 μL of CellTiter-Glo reagent were added directly into each well. The culture plate was shaken at 300 rpm for 5 min and then incubated at room temperature for 25 min to stabilize the luminescent signal. Luminescence was measured using the same microplate reader and normalized to number of cells.

### 2.8. 3D Migration Assays

To determine the effects of metformin on spheroid migration, spheroids were co-cultured CAFs using culture inserts (Greiner Bio-one, Kremsmünster, Austria). Fibroblasts were seeded in 12 well plates (1.5 × 10^5^ cells/well), and the next day, they were activated with conditioned media (see above) for 24 h to induce CAFs. At the same time, matrigel solution was prepared according to the manufacturer’s instructions. Briefly, Corning^®^ Matrigel^®^ Basement Membrane Matrix (BD Biosciences, Franklin Lakes, NJ, USA) was mixed with DMEM to obtain the final solution of 0.3% Matrigel. The solution was added to the growth surface of culture inserts (transparent membrane with 3 μm pores), and let it to be hydrated overnight at 37 °C. Then, spheroids were selected and individually laid on the matrigel in the inserts and placed in the multiwell plates with CAFs. As controls, spheroids in single culture (without CAFs in the bottom well), were used. In treated samples, 2 mM or 4 mM metformin was added to both insert and well. After 5 days, single cultured and co-cultured spheroids (treated or not with metformin), were fixed and stained with crystal violet. Images were acquired by AxioCam MRc digital camera for microscopy (AxioVision, Oberkochen, Germany). Migration areas were calculated as the difference between areas at day 0 and day 5 obtained by drawing circles around the spheroids at those times.

### 2.9. Immunofluorescent Spheroids Staining

The matrigel solution was added to chamber slides (Nunclon Sphera, Thermo Fisher Scientific, Waltham, MA, USA) and let to be hydrated for 1 h at 37 °C; then a single spheroid was laid on the matrigel. After 3 h, conditioned medium plus 2 mM or 4 mM of metformin was added. At day 5, spheroids were fixed with 4% paraformaldehyde for 30 min, and permeabilized with 0.1% Triton X-100 in PBS with for 45 min at room temperature. Spheroids were incubated overnight at 4 °C with phalloidin (Thermo Fisher Scientific, Waltham, MA, USA) to visualize actin filaments, and with SYTOX™ Green Nucleic Acid Stain (Invitrogen, Waltham, MA, USA) for nuclei staining. Confocal images were acquired by a Leica SP2-AOBS microscope, as follows. HCX PL APO 63 × 1.4 NA objective, voxel size x = 0.232 μm, y = 0.232 μm, z = 0.244 μm. Images were prepared for publication using the Fiji software [34] and are shown as maximum intensity projection along the *z*-axis.

### 2.10. Transwell Migration Assay

Fibroblast migration was assessed by 8 μm pore size transwell inserts (Greiner Bio-one). 5 × 10^4^ fibroblasts, in complete DMEM, were allowed to adhere to the membrane of the inserts overnight. The next day, complete DMEM was replaced with serum free medium plus or not (control) 4 mM metformin, while complete DMEM in the lower wells was used as chemoattractant. After 30 h at 37 °C, cells migrated to the lower side of the inserts were stained with crystal violet in 0.1% methanol and observed under the microscope. To obtain a quantitative analysis, inserts were eluted with DMSO and the dye present in the solution was quantified as OD at 560 nm by the spectrophotometer VICTOR^3^ 1420.

### 2.11. Wound-Healing Migration Assay

Fibroblasts at 100% confluence were scored with a sterile 10 μL micropipette tip, washed extensively with PBS, and treated with or without 2 mM or 4 mM metformin in complete DMEM. Images were acquired at T0, T16 and T30h to visualize wound incoming cells. Wound healings were calculated as the difference of the wound areas at T0 and T16 or T30h and expressed as percentages.

### 2.12. Statistical Analysis

Data analyses were performed by GraphPad Prism Version 8.3.0 for Windows (GraphPad Software). The statistical significance was evaluated by one-way ANOVA followed by Bonferroni’s or Tukey’s multiple comparison test or by *t*-test. A *p* value of less than 0.05 was considered significant.

## 3. Results

### 3.1. Effects of Metformin on MTT and Fibroblast Proliferation and Metabolism

In order to investigate the effect of metformin on proliferation of MTT wt, MTT SDHB sil, and primary fibroblast, we treated the cells with increasing doses of metformin (0, 2, 4, 8 mM) and followed their growth over time (one, two, three, and six days). Interestingly, we found that proliferation of MTT tumour cells (either wt or SDHB sil) was significantly decreased already after two days of treatment with 4 and 8 mM of metformin, as compared to their untreated counterparts. At day six, the proliferation was significantly reduced with all doses tested (Figure 1A).

In contrast, we observed a reduction in the growth of primary fibroblasts only after six days of treatment and only with the higher doses of metformin (4 mM and 8 mM) (Figure 1A), suggesting that fibroblasts could be less sensitive to metformin. Since after three days from the plating, we observed a flattening of the growth curve of fibroblasts (probably due to a contact inhibition), we decided to repeat the fibroblast count starting from a smaller number of initial cells and continuing the experiment up to nine days. Indeed, at this time, we observed a significant inhibition of fibroblast proliferation rate even with 2 mM of metformin treatment. The difference in drug responsiveness between MTT and fibroblasts was confirmed by evaluation of cyclin D1 expression by western blot. As shown in Figure 1B, in cells treated for three days with 4 mM metformin only MTT wt and SDHB sil, but not fibroblasts, showed a significant downregulation of cyclin D1, as compared with untreated cells.

In order to study metformin effects on cellular bioenergetics-considering its ability to inhibit the mitochondrial complex I activity-we quantified the intracellular concentration of ATP and lactate in all cell types. As expected, in the two MTT populations and in fibroblasts, 2 mM and 4 mM metformin for three days induced a significant dose-dependent decrease in ATP production (Figure 2A), with a parallel accumulation of intracellular lactate (Figure 2B).

We also observed that basal levels of ATP and lactate in untreated (control) fibroblasts were higher than those in MTT wt and MTT SDHB sil control cells, and they remained significantly higher in fibroblasts compared with MTT also after metformin treatment.

We also noted that the colour of culture media, following metformin treatment, turned into yellow, suggesting an acidification, most probably due to the increased levels of lactate production (Figure 3A).

Indeed, the media pH of all the cells treated with metformin, was significantly lower than pH of control media (Figure 3B) and the lowest pH was detected in fibroblast media. To figure out if the inhibition of proliferation was a consequence of the culture media acidification, MTT cells and fibroblasts were cultured in complete DMEM (pH 7.4) with or without metformin (4 mM), or in complete acidified DMEM (pH 6.5) for two, three, and six days, and then counted (Figure 3C). The results demonstrated that even at longer time (six days), only the treatment with metformin, but not with the acidified media, was able to induce a significant reduction of cell proliferation. Moreover, MTT proliferation was also significantly reduced when 4 mM of metformin was added in culture media containing 20 mM hepes, to buffer the acidification caused by drug treatment, demonstrating that the decrease in cell division rate, induced by metformin, it was pH independent (Figure 3D).

We also studied the growth in three-dimension (3D) of MTT wt and SDHB sil spheroids treated with metformin. Spheroids were cultured in 96-well round bottomed plates with or without 2 mM or 4 mM metformin, and their diameters were measured at different timepoints (two, three, and six days). We found that the growth of the 3D spheroids was already significantly reduced after two days of treatment with both 2 mM and 4 mM, with the exception of the 2 mM treated MTT SDHB sil spheroids in which it became significantly different on day three. In fact, diameters of both MTT wt and MTT SDHB sil treated spheroids remained mostly similar to those measured at T0, while growth of control spheroids was linear over time (Figure 4).

### 3.2. Metformin Inhibits Fibroblast, but Not MTT Spheroid Migration

We have previously shown that MTT spheroids co-cultured with CAFs significantly increased their migratory capability compared with single culture counterparts [27]. In fact, co-cultured spheroids developed long filamentous formations used by the cells to move and invade the surrounding matrix. Furthermore, the MTT SDHB sil cells not only exhibited more aggressive behavior than wt cells, but invaded the surrounding space by moving collectively, while the wt cells moved individually [27]. Therefore, we tested metformin effects on spheroid CAF-induced migration. We confirmed the inhibition of spheroid growth also in this set of experiments. As shown in Figure 5A,B, areas of single cultured spheroids treated with metformin were significantly lower than their untreated counterparts.

Surprisingly, migration of metformin-treated spheroids co-cultured with CAFs, was still evident, and migration areas were significantly higher compared with their metformin-treated, single-cultured, counterparts. Indeed, using the confocal microscopy, many cells detaching from co-cultured treated spheroids and invading the surrounding space were clearly detectable (Figure 5C). Furthermore, only in SDHB sil co-cultured spheroids the migration areas of the metformin-treated, were significantly different than those of the untreated co-cultured spheroids.

We next investigated whether metformin was able to inhibit fibroblast migration instead. We used two different approaches: the transwell assay, in which we stained the migrated cells (Figure 6A) and quantified the eluted dye by optical density (Figure 6B); and the wound healing assay, in which we scratched confluent fibroblasts and measured the healing of the wounds (Figure 6C,D).

Interestingly, in all cases, we found a significant reduction of fibroblast migration after metformin administration, with the exception of the 2 mM treatment at 16 h.

### 3.3. Metformin Has Different Outcomes on MTT and Fibroblast by Activating Different Pathways

We next investigated which pathways were regulated by metformin in the two MTT populations and in the fibroblasts. It is well known that metformin induces AMPK phosphorylation [13]. Western blot analyses showed that after 4 mM metformin, both MTT and fibroblasts significantly increased phosphorylation of AMPK at the Thr172 site, as compared to their untreated counterparts (Figure 7A).

It is worth noting, that fibroblasts showed a higher basal level of AMPK phosphorylation than tumour cells, and the increase of phosphorylation due to metformin is less apparent than that of MTT (Figure 7B). Following treatment with metformin phospho-Akt and phospho-Erk1/2 levels were significantly reduced only in MTT cells, but not in fibroblasts, which is consistent with the known role of Akt and ErK1/2 in supporting proliferation (Figure 7). We also explored the molecular mechanisms by which metformin might modulate cell migration in different cell types. 

EMT plays a major role in tumour progression, therefore we analysed the expression levels of N-cadherin and of E-cadherin, two well-known markers of EMT [35], the expression levels of these proteins were analysed (Figure 7). Interestingly, we found a significant downregulation of N-Cadherin with a concurrent upregulation of E-Cadherin in fibroblasts treated with metformin compared to untreated cells. A significant downregulation of N-Cadherin after metformin treatment was detectable only in MTT SDHB sil cells, but not in MTT wt. The absence of E-cadherin in MTT cells is not unexpected since chromaffin cells are nervous cells.

## 4. Discussion

In the present work, we have shown, for the first time, that metformin has different effects on cell growth and spread depending on the cell types present in PPGL TME. In particular, we found that metformin significantly inhibits the growth of MTT cells and spheroids, but surprisingly not their ability to migrate. Unexpectedly, metformin has an opposite effect on fibroblasts, by inhibiting their migration, but not proliferation, except for longer time of metformin treatment.

There are a few in-vitro studies performed on cell models derived from PPGL tumours that demonstrate an antiproliferative potential of metformin. Li and colleagues showed that metformin inhibited cell proliferation and promoted cell cycle arrest of pheochromocytoma rat-derived cells (PC12 cell line) [12]. Moreover, the authors found a downregulation of Cyclin-A2 (Ccna2) and Ccnb2 expression, which plays an important role in the regulation of cell cycle progression [12]. Analogous results on the inhibition of cell proliferation are also described in other two studies conducted on immortalized head and neck PGL cell cultures and again on PC12 cell line [13,36]. According with the aforementioned results, in this study we showed that, after metformin treatment, MTT cells undergo a decrease in proliferation and downregulate cyclin D1. This effect on proliferation was not observed in primary fibroblasts, with the except for the longest exposure at the higher doses of metformin, suggesting that these cells are less sensitive to metformin treatment. Erk1/2 and Akt pathways are mainly involved in the control of cell proliferation and are upregulated in several cancer types including PPGLs [37]. Recently, Li and co-authors [12] demonstrated a concomitant activation of AMPK and inhibition of Erk1/2 signalling pathways, upon metformin treatment. Consistently, we found a downregulation of Erk1/2 and Akt phosphorylation only in tumour cells but not in fibroblasts, supporting the antiproliferative effect of metformin only in the MTT cell lines.

It is widely accepted that transformed cells, growing in a monolayer on plastic dishes, have little in common with the complex 3D multicellular organization found in living organisms. Awareness of this discrepancy has led to studies intended to find more appropriate cellular models to better represent the in vivo scenario. These models include long-known multicellular spheroids, which seems to represent the best compromise. In fact, they can be easily standardized, and yet, they provide enough complexity, such as 3D geometry, to represent relevant aspects of human tumours [38]. Spheroids generation facilitates the growth of highly uniform spheroids and allow real-time monitoring of drug-induced changes in their phenotypes [27,39,40]. We previously demonstrated that TME, represented by primary CAFs, plays a fundamental and specific role in inducing the invasion properties of spheroid cells and that this effect is extraordinarily enhanced in MTT SDHB sil ones [27]. In the present work, we tested, for the first time, metformin effect on PPGL 3D culture, confirming metformin antiproliferative action also in this model. Remarkably, we observed only a partial reduction of the migration areas of co-cultured spheroids after metformin treatment, when compared to their co-cultured untreated counterpart. We believe that this reduction in the migration areas is due to a smaller cell number present in the spheroids as a result of the metformin treatment. It was also unexpected the antimigratory effect of metformin exerted on fibroblasts. Indeed, metformin targets fibroblasts in a different manner than PPGL cells, and this feature can be explained by the different modulation of intracellular signalling molecules. In fact, metformin shapes fibroblasts motility by downregulating N-cadherin expression and enhancing E-Cadherin expression. It has been reported that metformin plays a suppressor role in stroma-associated, inflammatory and fibrosis diseases such as renal fibrosis, cardiac fibrosis, and endometriosis [41]. Although benign diseases differ from neoplastic diseases, similar mechanisms have been corroborated to contribute to the stromal suppression in malignancies. In some cancers, metformin was demonstrated to repress the interstitial fibrosis [42]. Our findings on fibroblasts, components of TME, could explain metformin effects in regulating fibroblast migration and we hypothesized that metformin could model the phenotype of CAFs, resulting in a phenotype less prone to support tumour progression. Interestingly, metformin induced downregulation of N-Cadherin also in MTT SDHB sil cells, suggesting that the process of SDHB sil spheroid migration could be converted from a mesenchymal kind of migration (expressing high N-Cadherin levels) to an amoeboid and less aggressive migration type. It has previously proposed EMT as the process likely responsible for SDHB-related malignancy in PPGL [43]. Since chromaffin cells have non-epithelial origin, it would be more appropriate to use the term “neuroendocrine-mesenchymal transition”, as previously described in immortalized mouse chromaffin cells (imCC) [43]. Indeed, in MTT cells, the expression of E-cadherin-which is often used as a hallmark of EMT in epithelial cells-was barely detectable. However, in SDHB sil cells, we observed a downregulation of N-Cadherin after metformin administration.

Metabolically, we observed that in controls (metformin-untreated cells), basal ATP content was lower in MTT cells compared to that in fibroblasts. This phenomenon could be explained by Warburg’s observation that tumour cells show a metabolic shift from oxidative phosphorylation to aerobic glycolysis also in presence of oxygen [44,45]. Interestingly, lactate basal level was higher in fibroblasts than in MTT, suggesting that the metabolism of these cells is strictly dependent on glucose as nutrient. This could also justify the fact that fibroblasts produce more lactate after metformin treatment. This finding is supported also by our recent study [33] indicating that, when fibroblasts were kept in a low-glucose medium, intracellular ATP levels significantly decreased as compared to MTT cells, demonstrating that fibroblasts are more sensitive to glucose changes, while cancer cells have a greater plasticity and can likely use the lactate as an energy source. After metformin treatment, ATP concentrations were significantly decreased in all the cell types, suggesting the well-known inhibition of the mitochondrial complex I by this drug (for recent reviews see [46,47]). Moreover, while Meireles and colleagues [13] demonstrated a reduction in oxygen consumption and ATP production only when cells were treated at high dose of metformin (20 mM), in our experimental conditions this effect was obtained with a 5-fold lower dose. It is also well-known that the metformin-induced metabolic shift toward aerobic glycolysis induces an increase in lactate production [48]. In line with these findings, we also found a concomitant increase of lactate concentration both in tumour cells and fibroblasts after metformin addition. Most likely, to overcome the excessive intracellular lactate accumulation, this molecule could be released into the extracellular medium, causing its acidification. This hypothesis is supported by the observation that the maximum acidification of the culture medium was observed in fibroblast medium, and fibroblasts are in fact the cells that mostly accumulate lactate. Since the lowest medium pH found was 6.62 ± 0.06, we acidified the medium until a pH of 6.5, to verify whether medium acidification was sufficient to inhibit MTT cell proliferation, but this was not the case. This suggests that metformin *per se*, and not the acid pH, was responsible to growth inhibition.

It has been described that an imbalanced redox status, and a subsequent cellular metabolic stress ends with AMP-Activated Protein Kinase (AMPK) phosphorylation [31,32]. However, in our experimental models, despite MTT wt and SDHB sil cells showed lower basal levels of ATP compared with fibroblasts, basal phospho-AMPK was barely detectable. Nonetheless, in this paper we confirm that metformin induces an activation of phospho-AMPK both in tumour cells and in primary fibroblasts. It is reasonable to hypothesise that, since in fibroblasts phospho-AMPK is already high, the increase of AMPK phosphorylation in these cells is less evident than that observed in MTT cells.

PPGLs are mostly benign tumours, but when their metastatic transformation occurs, nowadays no efficacy treatment is available. Since the incidence of these tumours is so rare, at present there is no patient-related information available on the efficacy of metformin treatment on PPGLs. Herein, we provide evidence that metformin may represent a promising tool to impair TME homeostasis, since fibroblasts are fundamental regulators of cancer progression and invasion. Hence, targeting their biology could lead to develop complementary strategies to target tumour spread and progression.

## 5. Conclusions

Cell growth, but not migration of 3D PPGL spheroid cells is inhibited by metformin, which surprisingly exerted an opposite effect on fibroblasts, by inhibiting their migration, while their proliferation is only reduced after long-term exposure. These data indicate that different TME cell types respond differently to metformin treatment. This heterogeneity in the metformin action could be an effective strategy to synergistically target the tumour mass through a diverse approach, depending on the cell type. In summary, despite a previous emphasis of the effects of metformin on cancer cells, present findings show that adjacent stromal cells might represent a further target to allow PPGL tumour regression.

## Figures and Tables

**Figure 1 cancers-14-03471-f001:**
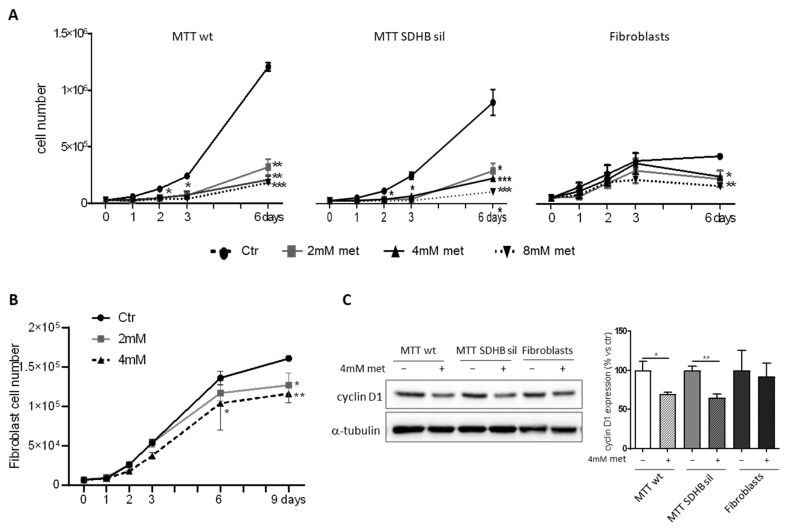
Analysis of metformin effects in terms of proliferation. (**A**) Proliferation of MTT wt, MTT SDHB sil, and fibroblasts was monitored at different timepoints following administration with 0, 2, 4, 8 mM doses of metformin. Proliferation of MTT tumour cells (wt and SDHB sil) was significantly decreased already after two days of metformin at 4 and 8 mM compared to their non-treated counterparts (* *p* < 0.05). This difference was maintained also at day three. After six days the proliferation of both MTT populations was significantly reduced with all the drug doses (2, 4, 8 mM, ** *p* < 0.01, *** *p* < 0.001). Fibroblasts were statistically lower only after long-term treatment (six days) with the higher doses of metformin (4 and 8 mM, * *p* < 0.05, ** *p* < 0.01, respectively). (**B**) Proliferation of fibroblasts was also evaluated at day nine, seeding 5 × 10^3^ cells/well. At this time, cell division was statistically decreased even with 2 mM of metformin (* *p* < 0.05). In both (**A**,**B**) graphs represent three independent experiments performed in duplicates. Asterisks indicate significance by two-way ANOVA with Bonferroni post hoc test ± SD. (**C**) Representative western blot of cyclin-D1 expression levels in MTT wt, MTT SDHB sil and fibroblasts. Alpha-tubulin immunoblot was used as loading control. Graph bars derived from the optical density analysis of western blot bands of three independent experiments ± SD. Asterisks indicate significance (* *p* < 0.05, ** *p* < 0.01, *** *p* < 0.001) by one tailed paired *t*-test.

**Figure 2 cancers-14-03471-f002:**
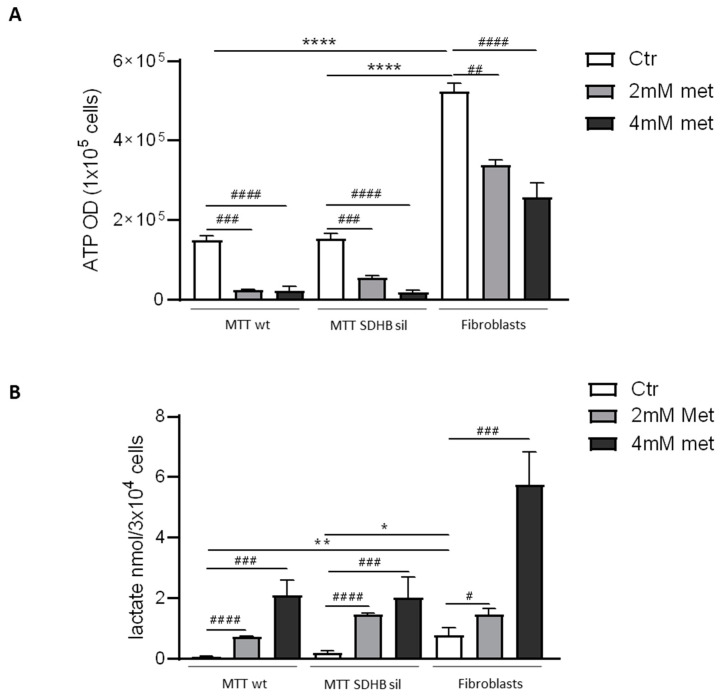
Effect of metformin on metabolic parameters. Intracellular ATP and lactate concentrations were measured in MTT wt, MTT SDHB sil and fibroblasts cells after 72 hours of 2 mM (light grey bars) or 4 mM metformin (dark grey bars) or 0 mM metformin (white bars) treatment and normalized on cell number. (**A**) A significative reduction of ATP level was detected in MTT wt and MTT SDHB sil cells, and in fibroblasts following both 2 and 4 mM of metformin treatment. The ATP basal levels in fibroblasts were significantly higher compared with both MTT cells (hashtags indicate the statistic of ATP values between treated and untreated cells, ^##^
*p* < 0.01, ^###^
*p* < 0.001, ^####^
*p* < 0.001. Asterisks indicate the statistic of ATP values among the different untreated cell populations, **** *p* < 0.0001). (**B**) A significative increase in intracellular lactate level was observed in all the cell populations after metformin treatment with both 2 mM and 4 mM (hashtags indicate the statistic of lactate values between treated and untreated cells, ^#^
*p* < 0.05, ^###^
*p* < 0.001, ^####^
*p* < 0.001. Asterisks indicate the statistic of lactate values among the different untreated cell populations, * *p* < 0.05, ** *p* < 0.01). Bars of both ATP and lactate are the means of four independent experiments with two replicates each ± SD. Hashtags and asterisks indicate significance by one tailed paired *t*-test.

**Figure 3 cancers-14-03471-f003:**
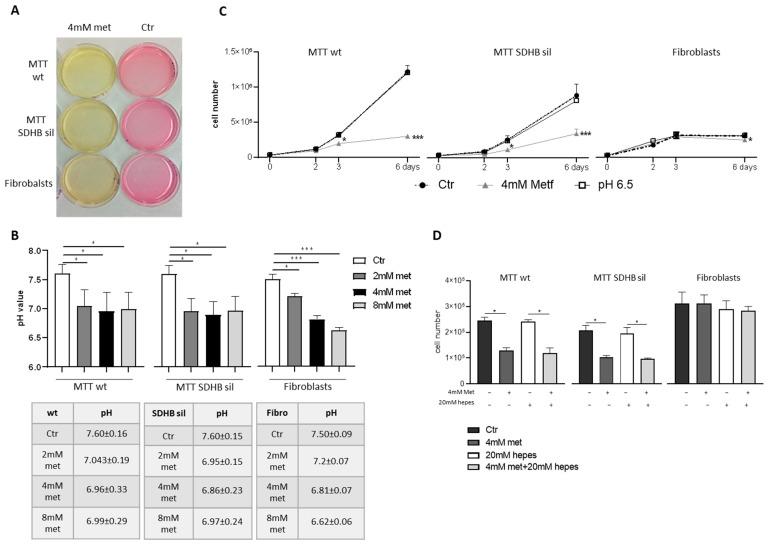
Effect of metformin in culture medium acidification. (**A**) Culture media turned into a yellow colour after 72 h of metformin administration. (**B**) pH measurement showed a significant acidification of the media of MTT (both wt and SDHB sil), and fibroblasts after 2 mM (dark grey bars), 4 mM (black bars) and 8 mM (light grey bars) metformin treatment, compared with not treaded controls (white bars). Bars are the media of three independent experiment ± SD. Asterisks indicate significance (* *p* < 0.05, *** *p* < 0.001) by one-way ANOVA with Dunnet post hoc test. Tables indicate the pH value ± SD. (**C**) Proliferation of MTT wt, MTT SDHB sil, and fibroblasts grown in acidic medium (pH 6.5) or in 4 mM metformin medium compared with controls (cells grown in DMEM + 10% FCS, pH 7.4). Graphs represent three independent experiments performed in duplicates, and asterisks indicate significance (* *p* < 0.05) by two-way ANOVA with Bonferroni post hoc test ± SD. (**D**) Cell proliferation of control cells (black bars), after 72 h of 4 mM metformin administration (dark grey bars), DMEM plus 20 mM of hepes (white bars), and following 20 mM of hepes plus 4 mM metformin 72 h (light grey bars). Graphs showed a significative reduction of only MTT growth after metformin treatment regardless of the presence of hepes in the medium, confirming that the effect of metformin was not acidic-dependent. Bars represent two independent experiments performed in triplicates, and asterisks indicate significance (* *p* < 0.05, *** *p* < 0.001) by one-way ANOVA with Tukey post hoc test ± SD.

**Figure 4 cancers-14-03471-f004:**
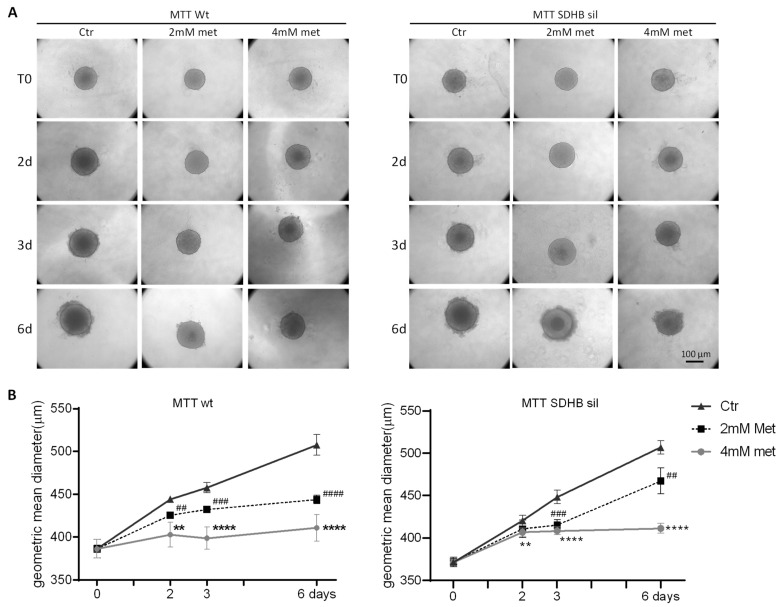
Metformin effects on 3D tumour spheroid growth. (**A**) Representative images of MTT spheroid growth (MTT wt on left, and MTT SDHB sil on right) at different timepoints, treated or not with 2 mM or 4 mM metformin. (**B**) Diameters of MTT wt (left panel) and MTT SDHB sil (right panel) spheroids were measured at two, three, and six days after 2 mM or 4 mM of metformin administration (black dotted lines and grey lines, respectively) compared to not treated controls (black lines). Graphs represent three independent experiments preformed in sixtuplicates. Asterisks indicate significance (** *p* < 0.01, **** *p* < 0.0001, ^##^ *p* < 0.01, ^###^ *p* < 0.001, ^####^ *p* < 0.0001) by two-way ANOVA with Bonferroni post hoc test ± SD.

**Figure 5 cancers-14-03471-f005:**
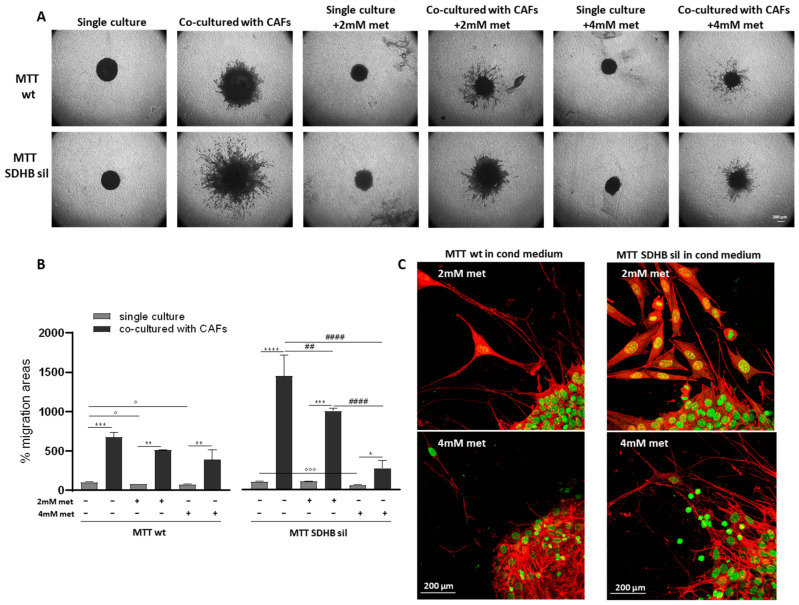
Effects of metformin on microenvironment-induced tumour spheroid migration. (**A**) Representative images of MTT wt and MTT SDHB sil spheroids, laid on 0.3% matrigel in transwell inserts, co-cultured or not with CAFs, and treated or not with 2 mM or 4 mM of metformin, acquired at low magnification (5×). Migration was observed after five days. In single culture, spheroid migrating cells were barely detectable. When spheroids were co-cultured with CAFs, plated in the lower compartment, a clear detachment of cells that invaded the surrounding space was observed. Spheroids in single cultures treated with 2 mM or 4 mM of metformin remained smaller compared with not treated spheroids, and in co-cultures, spheroid cell migration was only partially affected by metformin. Indeed, the core of the spheroid remained small, but migratory processes were still visible. (**B**) Bar graph quantifies the migration processes of MTT wt (left panel) and MTT SDHB sil (right panel) in single culture (light grey bars) on in co-culture with CAFs (dark grey bars) treated or not with 2 mM or 4 mM of metformin. Histograms represent the media of three independent experiments ± SD. Asterisks, hashtags, and circles indicate significance (Asterisks indicate the statistic of values between treated and untreated spheroids, * *p* < 0.05, ** *p* < 0.01, *** *p* < 0.001, **** *p* < 0.0001. Hashtags indicate the statistic of values among the different treated spheroid populations, ^##^
*p* < 0.01, ^####^
*p* < 0.0001. Circles indicate the statistic of values among treated and untreated single cultured spheroids, ° *p* < 0.05, °°° *p* < 0.001) by one-way ANOVA with Tukey’s multiple comparisons post hoc test. (**C**) Representative images of tumour spheroid migration acquired by confocal microscopy at high magnification (63×). Spheroids, grown in CAFs conditioned medium and 2 mM or 4 mM of metformin, were labelled with phalloidin (Thermo Fisher Scientific) to visualize actin filaments (in red), and with SYTOX™ Green Nucleic Acid Stain (Invitrogen) for nuclei staining (in green). Confocal images were acquired with a Leica SP2-AOBS, processed using Fiji software, and shown as maximum intensity projection along the *z*-axis. After five days, the elongation processes were still present in both MTT wt and MTT SDHB sil spheroids.

**Figure 6 cancers-14-03471-f006:**
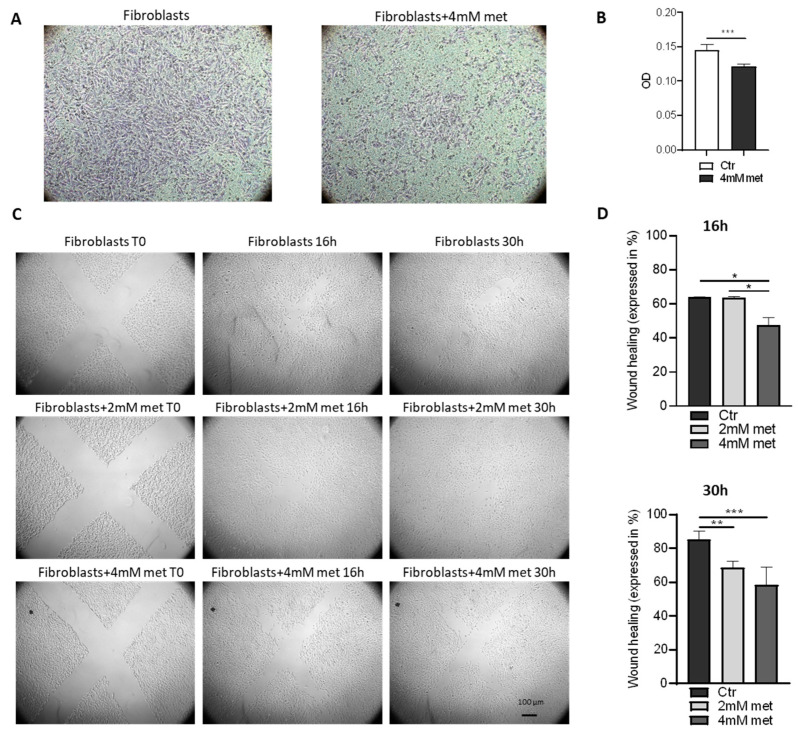
Effects of metformin on fibroblasts migration. (**A**) Fibroblast migration through 8µm porous transwell membranes in presence or not of 4 mM metformin for 30 h. Images are representative of migrating cells of three independent experiments, each performed in duplicate. (**B**) Quantification of fibroblast migration reported as optical density of crystal violet measurements. Fibroblasts treated with 4 mM of metformin (dark grey bar) migrated significantly less compared with their not treated counterpart (white bar). (**C**) Representative images of wound healing assay of fibroblasts treated or not with 2 mM or 4 mM of metformin. Images are representative of three independent experiments, each performed in triplicate. The efficiency of the healing was evaluated after 16 h and 30 h. (**D**) Histograms quantify the healing of the scratch areas by fibroblasts treated or not (black bars) with 2 mM (light grey bars) or with 4 mM of metformin (dark grey bars). Bar charts in (**B**,**D**) represent the media of three independent experiments ± SD. Asterisks indicate significance (* *p* < 0.05, ** *p* < 0.01; *** *p* < 0.001) by one tailed paired *t*-test.

**Figure 7 cancers-14-03471-f007:**
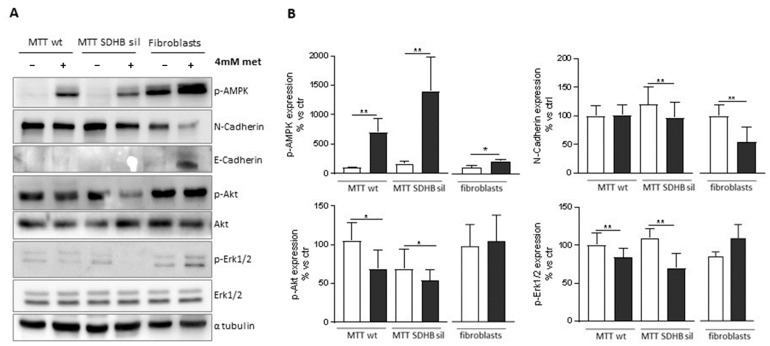
Effect of metformin on activation of intracellular signalling pathways in MTT and fibroblast cells. (**A**) Representative western blot of expression levels of Phospho-AMPK, Phospho-AKT, AKT, Phospho-ERK1/2, ERK1/2, N-Cadherin and E-Cadherin in MTT wt, MTT SDHB sil and fibroblasts treated or not with 4 mM of metformin for 72 h. α-tubulin immunoblot was used as loading control. (**B**) Densitometric analysis of Phospho-AMPK, Phospho-AKT, AKT, Phospho-ERK1/2, ERK1/2, N-Cadherin and E-Cadherin in MTT wt, MTT SDHB sil and fibroblasts treated (dark grey bars) or not (white bars) with 4 mM of metformin for 72 h. Graph bars indicate the mean of three independent experiments ± SD. Asterisks indicate significance (* *p* < 0.01; ** *p* < 0.01) by one tailed paired *t*-test.

## Data Availability

The data presented in this study are available in the article and Appendix A.

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
