# Peer review of "Metformin Treatment Induces Different Response in Pheochromocytoma/Paraganglioma Tumour Cells and in Primary Fibroblasts"

_cancers, 2022, doi:10.3390/cancers14143471_

Round 1

Reviewer 1 Report

The authors answered to all my concerns.

Author Response

The Authors thank once again the Reviewer

Reviewer 2 Report

The authors replied all of the previous comments.

Author Response

The Authors thank the Reviewer

Reviewer 3 Report

In the abstract the authors state: “On the other hand, metformin had no effect on proliferation of mouse primary fibroblasts…”. In the results section, on the other hand: “we decided to repeat the fibroblast count starting from a smaller number of initial cells and continuing the experiment up to 9 days. Indeed, at this time, we observed a significant inhibition of fibroblast proliferation rate even with 2mM of metformin treatment”. The, again, in the discussion: “Unexpectedly, metformin has an opposite effect on fibroblasts, by inhibiting their migration, but not proliferation”. This is then repeated under the Conclusions. These statements need to be harmonized, perhaps starting from the results where a significant inhibition of the proliferation is now demonstrated.

Author Response

The Authors thank the Reviewer for the suggestion. The Authors modified the sentences trying to harmonise them.

This manuscript is a resubmission of an earlier submission. The following is a list of the peer review reports and author responses from that submission.

Round 1

Reviewer 1 Report

Martinelli et al. report the effects of Metformin on a pheochromocytoma mouse cell line.  Others publications had previously reported same effects on cells proliferation and/or cells migration as in this article (Khallaghi et al. 2016, Meireles et al. 2022, li et al. 2017…).

This present study has some limitations:

Major points

- The different migration assays were performed on 30 hours. However on this duration authors cannot exclude that cells proliferation is involved. This duration induce a major bias on the study on the migration evaluation.

- Why do the authors choose 4 mM of Metformin to perform most of the experiment whereas 2 mM seems to work on proliferation? It will be very interesting to have the main results with 2 mM.

- Medium acidification was described to have effect on cellular proliferation. It will be interesting to have proliferation and migration assay on cells treated by Metformin with a less acid medium (medium with a buffer like hepes for example).

Minor points

In the introduction section the authors wrote, “Moreover, almost 80 % of SDHB mutated patients develop metastatic lesions (Andrews et al)”. In Andrews et al. approximately 20 % of SDHB proband and less than 5 % of non-proband develop a metastatic disease.

Author Response

Dear Review,

               the Authors wish to thank all the Reviewers, who gave them the opportunity to improve and refine the paper.  The Authors submit a revised version according to Reviewers’ suggestions. All changes are specified in the answer to the Reviewers and highlighted in yellow in the manuscript.

REVIEWER 1

Martinelli et al. report the effects of Metformin on a pheochromocytoma mouse cell line.  Others publications had previously reported same effects on cells proliferation and/or cells migration as in this article (Khallaghi et al. 2016, Meireles et al. 2022, li et al. 2017…).

This present study has some limitations:

Major points

- The different migration assays were performed on 30 hours. However on this duration authors cannot exclude that cells proliferation is involved. This duration induce a major bias on the study on the migration evaluation.

The Authors thank the reviewer for the suggestion. The wound healing migration assay was also performed at 16h, and the results are now reported in Fig. 6 C and D. Even at 16h, fibroblasts treated with 4mM of metformin migrated significantly less than their not treated counterparts. It is worth noting that fibroblast proliferation was not affected by metformin treatment during the first 3 days, the Authors believe that the differences in migration were most likely due to the treatment with metformin.    

- Why do the authors choose 4 mM of Metformin to perform most of the experiment whereas 2 mM seems to work on proliferation? It will be very interesting to have the main results with 2 mM.

At the beginning of the project the Authors performed most of the experiments with different doses of metformin, then they decided to show mainly the results obtained after 4mM treatment. Following the Reviewer’s suggestion, in the revised version of the manuscript the Authors also included the main results obtained with 2 mM. which were already available in the lab. In particular, ATP and lactate levels (Fig. 2A and B, respectively), decreased of pH media (Fig. 3B), 3D tumour spheroid growth (Fig. 4), spheroid migration (Fig. 5), and fibroblast wound healing assay (Fig. 6C and D) have now been included in the paper.

- Medium acidification was described to have effect on cellular proliferation. It will be interesting to have proliferation and migration assay on cells treated by Metformin with a less acid medium (medium with a buffer like hepes for example).

The Authors performed the cell proliferation experiments adding 20mM of hepes, or 20mM of hepes plus 4mM metformin to the culturing media for 72h. The results are now included in Fig. 3D. The bar graph showed a significative reduction of MTT growth after metformin treatment regardless to the presence of hepes in the medium, confirming that metformin effect was not acidic-dependent.

Minor points

In the introduction section the authors wrote, “Moreover, almost 80% of SDHB mutated patients develop metastatic lesions (Andrews et al)”. In Andrews et al. approximately 20 % of SDHB proband and less than 5 % of non-proband develop a metastatic disease.

The Authors thank once again the Reviewer, it was a typo mistake.

Reviewer 2 Report

This manuscript by Martinelli et. al. investigated the different reaction of MTT cell line and primary fibroblasts to metformin. They demonstrate metformin inhibited the proliferation of MTT both in 2D and 3D cell culture, while fibroblast was not inhibited. On the other hand, metformin inhibited the migration and invasion ability of fibroblast, whereas migration ability of MTT was not affect by metformin. Although it had been shown that metformin inhibit the proliferation and viability of PC12, a pheochromocytoma cell line derived from rat, this manuscript provided a novel aspect that metformin may affect the tumour microenvironments. Overall, the authors present interesting preclinical data on the antitumour ability of metformin, but there is yet a lack of robust clinical-translational data at this time.

Specific questions/comments:

1.       Since 2017, the terms benign and malignant PPGL were replaced by non- and metastatic PPGL.

2.       Figure 1B, the authors may put the WB figures aside the bar chart to save the space.

3.       The sentence below ‘In contrast, we observed a reduction in the growth of primary fibroblasts only after 6 days of treatment and only with the highest doses of metformin (4 and 8mM) (Fig. 1A)’. Two ‘highest’ dose?

4.       Figure 2, there are too many asterisks which is confusing, the authors maybe use different symbols to show distinguished comparison. For example, use # to show the comparison between Ctrl and Met treatment.

5.       Figure 5, did you also test the effect of metformin on migration ability for MTT with 2D culture?

Figure 5B, the legend indicated (upper panel) and (lower panel), which is not consist with the figure.

6.       Met showed an inhibition of MTT migration, as shown in figure 5B, but the authors declare not inhibit the migration ability in the first paragraph of discussion part. As the authors discussed in the fifth paragraph of discussion part, ’We believe that this reduction in the migration areas is due to a smaller cell number present in the spheroids because of the metformin treatment’. However, figure 4 didn’t show a smaller size of tumour after Met treatment.

7.       The discussion on the different reaction of MTT and fibroblast is an important part and associated more with the title, maybe should be discussed earlier rather than the discussion part for ATP and PH.

8.       The authors showed a different regulation of P-ERK/P-AMPK/N-cadherin by metformin, accompanied with the inhibition of proliferation/migration ability on cells. However, these is not enough to conduct a conclusion ‘Metformin, via activation of the AMPK pathway, inhibited cell growth of PPGLs cells, as well as of tumour 3D PPGL spheroids, by downregulation of phospho-Akt and phospho-Erk1/2 expression.’

Author Response

Dear Review,

               the Authors wish to thank all the Reviewers, who gave them the opportunity to improve and refine the paper.  The Authors submit a revised version according to Reviewers’ suggestions. All changes are specified in the answer to the Reviewers and highlighted in yellow in the manuscript.

REVIEWER 2

This manuscript by Martinelli et. al. investigated the different reaction of MTT cell line and primary fibroblasts to metformin. They demonstrate metformin inhibited the proliferation of MTT both in 2D and 3D cell culture, while fibroblast was not inhibited. On the other hand, metformin inhibited the migration and invasion ability of fibroblast, whereas migration ability of MTT was not affect by metformin. Although it had been shown that metformin inhibit the proliferation and viability of PC12, a pheochromocytoma cell line derived from rat, this manuscript provided a novel aspect that metformin may affect the tumour microenvironments. Overall, the authors present interesting preclinical data on the antitumour ability of metformin, but there is yet a lack of robust clinical-translational data at this time.

Specific questions/comments:

  1. Since 2017, the terms benign and malignant PPGL were replaced by non- and metastatic PPGL.

The Authors thank the Reviewer for the observation. The term benign in the text has now been substituted with non-metastatic.

  1. Figure 1B, the authors may put the WB figures aside the bar chart to save the space.

The WB figures are now aside the bar chart.

  1. The sentence below ‘In contrast, we observed a reduction in the growth of primary fibroblasts only after 6 days of treatment and only with the highest doses of metformin (4 and 8mM) (Fig. 1A)’. Two ‘highest’ dose?

The Authors corrected the sentence substituting “highest doses” with “higher doses”.

  1. Figure 2, there are too many asterisks which is confusing, the authors maybe use different symbols to show distinguished comparison. For example, use # to show the comparison between Ctrl and Met treatment.

The Authors thank the Reviewer for the suggestion, in the present revised version different symbols to show distinguished comparison have now been used.

  1. Figure 5, did you also test the effect of metformin on migration ability for MTT with 2D culture?

The Authors performed some experiments to test the effect of metformin on MTT migration ability in 2D culture, but they encountered some technical problems, mainly due to the fact that MTT grow in clusters. The results obtained strongly suggested a lack of an inhibitory effect of metformin on migration, but the assay was not so clean and the authors preferred to show only the 3D results which however better resemble the in vivo conditions.

            Figure 5B, the legend indicated (upper panel) and (lower panel), which is not consist with the figure.

Upper panel and lower panel have now been corrected with left panel and right panel.

  1. Met showed an inhibition of MTT migration, as shown in figure 5B, but the authors declare not inhibit the migration ability in the first paragraph of discussion part. As the authors discussed in the fifth paragraph of discussion part, ’We believe that this reduction in the migration areas is due to a smaller cell number present in the spheroids because of the metformin treatment’. However, figure 4 didn’t show a smaller size of tumour after Met treatment.

In the figure 4, the Authors demonstrated that the sizes of the spheroids treated with metformin are significantly smaller than the sizes of not treated ones. The Authors also believe that the reduction in the migration areas was due to a smaller cell number forming the spheroids.

  1. The discussion on the different reaction of MTT and fibroblast is an important part and associated more with the title, maybe should be discussed earlier rather than the discussion part for ATP and pH.

The different reaction of MTT and fibroblast is now discussed earlier in the discussion session.

  1. The authors showed a different regulation of P-ERK/P-AMPK/N-cadherin by metformin, accompanied with the inhibition of proliferation/migration ability on cells. However, these is not enough to conduct a conclusion ‘Metformin, via activation of the AMPK pathway, inhibited cell growth of PPGLs cells, as well as of tumour 3D PPGL spheroids, by downregulation of phospho-Akt and phospho-Erk1/2 expression.’

The Authors thank the Reviewer for the correct observation. They rewrote the first sentence of the conclusion.

Reviewer 3 Report

There are the following points of concern that the authors should address:

- The authors should explain why they used the doses of 2, 4 and 8 mM metformin. Did they do so on the basis of literature data or of preliminary experiments that they had performed?

- The difference in sensitivity to metfromin between MTTwt and MTT SDHB sil on one hand and fibroblasts is not very convincing. In fact, looking at Fig. 1A one is led to argue that the difference is simply due to the fact that the MTT cells have a much shorter doubling time than fibroblasts. In fact, the effect of metformin on MTT cells on day 3 and the effect of metformin on fibroblasts on day 6 appear very similar. What would the picture for fibroblasts look like if, for example, one would measure proliferation on day 9 or 12 (using, at the beginning, an appropriate cell seeding, obviously)? To this regard, and in spite of the claimed lack of effect of metformin on fibroblast proliferation, in the discussion the authors state (page 14 of 16): It was also unexpected the antiproliferative effect of metformin exerted on fibroblasts. This is clearly contradictory to what they wrote before. I presume this is a mistake

- In the summary the authors claim that the fibroblasts exhibited a downregulation of N-cadherin and an upregulation of E-cadherin in response to metformin. Later in the text, however, they show that MTTs undergo the same changes in response to metformin. For this reason, the suggestion that an epithelial (-like)-mesenchymal transition (partial or complete?) may underlie the differential response in terms of migration of the two cell types is not tenable.

- Overall, English language is OK but there are some typos that should be amended.

Author Response

Dear Review,

               the Authors wish to thank all the Reviewers, who gave them the opportunity to improve and refine the paper.  The Authors submit a revised version according to Reviewers’ suggestions. All changes are specified in the answer to the Reviewers and highlighted in yellow in the manuscript.

REVIEWER 3

There are the following points of concern that the authors should address:

- The authors should explain why they used the doses of 2, 4 and 8 mM metformin. Did they do so on the basis of literature data or of preliminary experiments that they had performed?

At the beginning of the project the Authors used for several experiments many different metformin concentrations, based on what found in the literature. Then Authors decided to use only the 2, 4, and 8mM doses, since they already affected cell metabolism and functions. Many main results obtained with the different doses were already available in the lab, and some of them with the 2mM one are now included in the paper as requested by the Reviewer 1.

- The difference in sensitivity to metfromin between MTTwt and MTT SDHB sil on one hand and fibroblasts is not very convincing. In fact, looking at Fig. 1A one is led to argue that the difference is simply due to the fact that the MTT cells have a much shorter doubling time than fibroblasts. In fact, the effect of metformin on MTT cells on day 3 and the effect of metformin on fibroblasts on day 6 appear very similar. What would the picture for fibroblasts look like if, for example, one would measure proliferation on day 9 or 12 (using, at the beginning, an appropriate cell seeding, obviously)?

In reality, the doubling time of fibroblasts is even higher than MTT during the first three days, as shown in figure 1A. Then the proliferation curve reaches a plateau, probably due to contact inhibition, since fibroblasts are not a cell line, but they have been obtained by primary cultures and they are not immortalised. Thanks to the Reviewer’s suggestion, the Authors chose an appropriate cell seeding, decreasing the initial number of cells, and repeated the proliferation experiments, counting fibroblasts up to day 9. The results obtained have now been reported in Fig. 1B.

To this regard, and in spite of the claimed lack of effect of metformin on fibroblast proliferation, in the discussion the authors state (page 14 of 16): It was also unexpected the antiproliferative effect of metformin exerted on fibroblasts. This is clearly contradictory to what they wrote before. I presume this is a mistake

The Authors are very sorry for this, which was obviously a mistake. It has now been corrected.

- In the summary the authors claim that the fibroblasts exhibited a downregulation of N-cadherin and an upregulation of E-cadherin in response to metformin. Later in the text, however, they show that MTTs undergo the same changes in response to metformin. For this reason, the suggestion that an epithelial (-like)-mesenchymal transition (partial or complete?) may underlie the differential response in terms of migration of the two cell types is not tenable.

MTT are nervous cells, and normally express high levels of N-cadherin, which is why it is not possible to refer to an epithelial-mesenchymal transition considering MTT cells. This is the reason why the Authors did not consider the change of N-cadherin observed in fibroblasts comparable, in terms of biological consequences, to that observed in MTT cells. Especially considering that this change was evident only in MTT SDHB sil cells. For these reasons, the Authors did not want to underline this aspect in the summary, and merely made some hypotheses in the discussion.

- Overall, English language is OK but there are some typos that should be amended.

The paper was read by a native English speaker. Hopefully some typos have now been corrected.

Reviewer 4 Report

Martinelli et al presents interesting and novel results of how manipulation of the microenvironment of tumour cells is affected by metformin. This is a well-written manuscript that adds to the scientific knowledge about the mechanisms of PPG tumour behaviour and potential targets of treatment. 

Author Response

The Authors wish to thank the Reviewer for the kind comments.